# NAS-Perf-360: Benchmarking Diverse Tasks for Neural Architecture Search

## Abstract

Most existing neural architecture search (NAS) benchmarks and algorithms prioritize performance on well-studied tasks, e.g., image classification on CIFAR and ImageNet. This makes the applicability of NAS approaches in more diverse areas inadequately understood. In this paper, we present **NAS-Perf-360**, a performance benchmark suite for state-of-the-art NAS methods for convolutional neural networks (CNNs).[1] To construct it, we curate a collection of ten tasks spanning a diverse array of application domains, dataset sizes, problem dimensionalities, and learning objectives. By carefully selecting tasks that can both interoperate with modern CNN-based search methods but that are also far-afield from their original development domain, we can use NAS-Perf-360 to investigate the following central question: *do existing state-of-the-art NAS methods perform well on diverse tasks?* Our experiments show that a modern NAS procedure designed for image classification can indeed find good architectures for tasks with other dimensionalities and learning objectives; however, the same method struggles against more task-specific methods and performs catastrophically poorly on classification in non-vision domains. The case for NAS robustness becomes even more dire in a resource-constrained setting, where a recent NAS method provides little-to-no benefit over much simpler baselines. These results demonstrate the need for a performance benchmark such as NAS-Perf-360 to help develop NAS approaches that work well on a variety of tasks, a crucial component of a truly robust and automated pipeline. We conclude with a demonstration of the kind of future research our suite of tasks will enable. All data and code is made publicly available.

## 1 Introduction

Neural architecture search (NAS) aims to automate the design of deep neural networks, ensuring performance on par with hand-crafted architectures while reducing human labor devoted to tedious architecture tuning (Elsken et al., 2019). With the growing number of application areas of ML, and thus of use-cases for automating it, NAS has experienced an intense amount of study, with significant progress in search space design (Zoph et al., 2018; Liu et al., 2019b; Cai et al., 2019), search efficiency (Pham et al., 2018), and search algorithms (Xu et al., 2020; Li et al., 2021a; White et al., 2021). While the use of NAS techniques may be especially impactful in under-explored or under-resourced domains where less expert help is available, the field has largely been dominated by methods designed for and evaluated on benchmarks in computer vision (Liu et al., 2019b; Ying et al., 2019; Dong & Yang, 2020). There have been a few recent efforts to diversify these benchmarks to settings such as vision-based transfer learning (Duan et al., 2021) and speech and language processing Mehrotra et al. (2021); Klyuchnikov et al. (2020); however, evaluating NAS methods on such well-studied tasks using traditional CNN search spaces does not give a good indication of their utility on more far-afield applications, which have often necessitated the design of custom neural operations (Cohen et al., 2018; Li et al., 2021b).

We make progress towards studying NAS on more diverse tasks by introducing a suite of benchmark datasets drawn from various data domains that we collectively call **NAS-Perf-360**. This benchmark consists of an organized setup of ten suitable datasets that (a) can be evaluated in a unified way using existing NAS approaches and (b) represent diverse application domains, dataset sizes, problem

---

[1] In this work, *NAS method* refers to a combined search space and algorithm pair, not the algorithm alone.

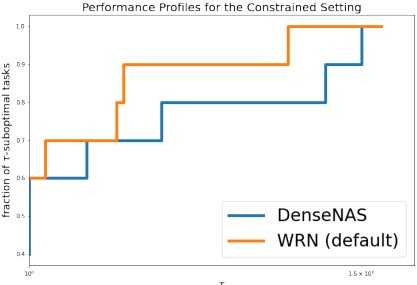 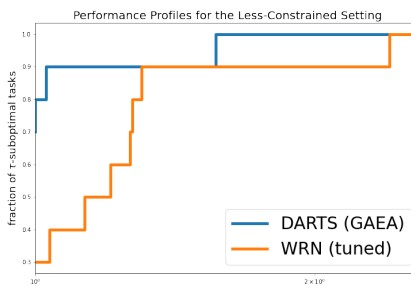

Figure 1: Performance profiles for two settings on all ten tasks in NAS-Perf-360. The y-value indicates the fraction of tasks on which a plotted method's error is within a multiplicative factor $\tau$ of the lowest error achieved by all plotted methods..

dimensionalities, and learning objectives. We also include standard image classification evaluations as a baseline point of comparison, as many new methods continue to be designed for such tasks.

Following our construction of this suite of tasks, we demonstrate both the usefulness of and need for NAS-Perf-360 by using it to investigate whether modern NAS is useful to practitioners faced with diverse tasks, i.e., whether its success in computer vision is indicative of strong performance on the much broader set of problems to which NAS can conceivably be applied. To address this question, we start with the fact that a common first approach when applying deep learning to a new domain is to try an off-the-shelf CNN; in our case, this will be the Wide ResNet (WRN) (Zagoruyko & Komodakis, 2016). We then consider the scenario of two practitioners: one with only the resources to train one WRN using the default settings and another that has enough to tune WRN using an off-the-shelf hyperparameter optimizer (Li et al., 2018). Both are faced with a decision: *should they use these fixed-architecture baselines or try out the best NAS has to offer?*

Overall, our empirical investigation suggests the following:

1. The less-constrained practitioner might usually do better using NAS—20% relative improvement over WRN on the median task—but risks catastrophic results on specific non-vision applications.
2. The robustness of NAS in the constrained case may be worse: the practitioner is likely better-off simply using the simple off-the-shelf WRN, as its median rank across NAS-Perf-360's ten tasks is the same as that of our candidate NAS method.

These results are obtained via experiments using two well-studied modern search spaces: the cell-based DARTS space (Liu et al., 2019b) and the efficiency-focused DenseNAS space (Fang et al., 2020). Each space is paired with a search method known to find well-performing architectures on ImageNet, specifically the state-of-the-art GAEA PC-DARTS (Li et al., 2021a) for the former and the original weight-sharing algorithm used by DenseNAS for the latter. Note that our assessment includes a more holistic comparison using performance profiles (c.f. Figure 1) to reinforce these ranking-based comparisons, which are useful but can miss a lack of robustness or exaggerate minor differences between methods.

The initial experimental results enabled by NAS-Perf-360 suggest that the robustness of modern search methods to diverse tasks beyond image classification is mixed at best. At the same time, our set of tasks can serve as a crucial tool for investigating and rectifying this issue, and it is thus important for moving towards a truly automated pipeline containing NAS. In particular, NAS-Perf-360 will facilitate such progress via a diverse array of tasks for validating NAS methods that are not only challenging, real-life problem settings but also computationally accessible for academic researchers with limited budgets. We demonstrate this potential via further studies on the comparative importance of search spaces v. search algorithms and the usefulness of more-customized approaches, specifically by studying a random search (RS) baseline over the DenseNAS space as well as two domain-specific methods: Auto-DeepLab (Auto-DL)(Liu et al., 2019a) for dense prediction and AMBER (Zhang et al., 2021b) for prediction from 1D data. Among other insights, these experiments provide evidence that a more robust NAS may require better search spaces with a wider variety of operations.

The associated datasets and experiment code will remain open-source and accessible at a temporarily anonymized repository `https://anonymous.4open.science/r/NAS-Bench-360-26D1`. Reproducibility of all experiments is assured from open-sourcing all relevant code for the end-to-end procedure, with Docker containers and random seeds provided.

## 2 RELATED WORK

Benchmarks have been critical to the development of NAS in recent years. This includes standard evaluation datasets and protocols, of which the most popular are the CIFAR-10 and ImageNet routines used by DARTS (Liu et al., 2019b). Another important type of benchmark has been tabular benchmarks such as NAS-Bench-101 (Ying et al., 2019), NAS-Bench-201 (Dong & Yang, 2020), and NAS-Bench-1Shot1 (Zela et al., 2020); these benchmarks exhaustively evaluate all architectures in their search spaces, which is made computationally feasible by defining simple searched cells. Consequently, they are less expressive than the DARTS cell (Liu et al., 2019b), often regarded as the most powerful search space in the cell-based regime. Notably, our benchmark is *not* a tabular benchmark, i.e., we do *not* evaluate every architecture from a fixed search space; instead, the focus is on the organization of a suite of tasks for assessing both NAS algorithms and search spaces, which would necessarily be restricted by fixing a search space for a tabular benchmark. Pre-computing on an expansive search space such as DARTS, with $10^{18}$ possible architectures, is computationally intractable. Architectures found on lesser search spaces are most likely suboptimal: the vanilla WRN outperforms all networks in the NAS-Bench-201 search space on CIFAR-100.

While NAS methods and benchmarks have generally been focused on computer vision, recent work such as AutoML-Zero (Real et al., 2020) and XD-operations (Roberts et al., 2021) has started moving towards a more generically applicable set of tools for AutoML. However, even more recent benchmarks that do go beyond the most popular vision datasets have continued to focus on well-studied tasks, including vision-based transfer learning (Duan et al., 2021), speech recognition (Mehrotra et al., 2021), and natural language processing (Klyuchnikov et al., 2020). We aim to go beyond such areas to evaluate the potential of NAS to automate the application of ML in truly under-explored domains. One analogous work to ours in the field of meta-learning is the Meta-Dataset benchmark of few-shot tasks (Triantafillou et al., 2020), which similarly aimed to establish a wide-ranging set of evaluations for that field. For our inclusion of diverse tasks, we title our benchmark NAS-Perf-360 to resemble the idea of a 360-degree camera that covers all possible directions.

## 3 NAS-PERF-360: A SUITE OF DIVERSE AND PRACTICAL TASKS

In this section, we introduce the NAS setting targeted by our benchmark, our motivation for organizing a new set of diverse tasks as a NAS evaluation suite, and our task-selection methodology. We report evaluations of specific algorithms on this new benchmark in the next section.

### 3.1 NEURAL ARCHITECTURE SEARCH: PROBLEM FORMULATION AND BASELINES

For completeness and clarity, we first formally discuss the architecture search problem itself, starting with the extended hypothesis class formulation Li et al. (2021a). Here the goal is to use a dataset of points $x \in \mathcal{X}$ to find parameters $\mathbf{w} \in \mathcal{W}$ and $a \in \mathcal{A}$ of a parameterized function $f_{\mathbf{w},a} : \mathcal{X} \mapsto \mathbb{R}_{\geq 0}$ that minimize the expectation $\mathbb{E}_{x \sim \mathcal{D}} f_{\mathbf{w},a}(x)$ for some test distribution $\mathcal{D}$ over $\mathcal{X}$; here $\mathcal{X}$ is the input space, $\mathcal{W}$ is the space of model weights, and $\mathcal{A}$ is the set of architectures. For generality, we do not require the training points to be drawn from $\mathcal{D}$ to allow for domain adaptation, as is the case for one of our tasks, and we do not require the loss to be supervised. Note also that the goal here does not depend on computational or memory efficiency, which we do not focus on in our evaluations; our restriction is only that the entire pipeline can be run on an NVIDIA V100 GPU.

Notably, this formulation makes no distinction between the model weights $\mathbf{w}$ and architectures $a$, treating both as parameters of a larger model. Indeed, the goal of NAS may be seen as similar to model design, except now we include the design of an (often-discrete) *architecture space* $\mathcal{A}$ such that it is easy to find an architecture $a \in \mathcal{A}$ and model weights $\mathbf{w} \in \mathcal{W}$ whose test loss $\mathbb{E}_{\mathcal{D}} f_{\mathbf{w},a}$ is low using a search algorithm. This can be done in a one-shot manner—simultaneously optimizing $a$ and

**w**—or using the standard approach of first finding an architecture $a$ and then keeping it fixed while training model weights **w** using a pre-specified algorithm such as stochastic gradient descent (SGD).

This formulation also includes non-NAS methods by allowing the architecture search space to be a singleton. When the sole architecture is a standard and common network such as WRN (Zagoruyko & Komodakis, 2016), this yields a natural baseline with an algorithm searching for training hyperparameters, not architectures. On the other hand, any architecture space $\mathcal{A}$ allows for non-one-shot methods to search for architectures, such as random search through repeatedly sampling architectures and evaluating them from partial training. We adopt this simple method as our random baseline. For our empirical investigation, we compare the performance of state-of-the-art NAS approaches against that of the two baselines.

## 3.2 TASK SELECTION: MOTIVATION AND METHODOLOGY

Curating a diverse, practical set of tasks for the study of NAS is our primary motivation behind this work. We observe that past NAS benchmarks focused on creating larger search spaces and more sophisticated search methods for neural networks. However, the utility of these search spaces and methods are only evaluated on canonical computer vision datasets. On a broader range of problems, whether these new methods can improve upon simple baselines remains an open question. This calls for the introduction of new datasets lest NAS research overfits to the biases of CIFAR-10 and ImageNet. By identifying these possible biases, future directions in NAS research can be better primed to suit the needs of practitioners and to increase the deployment of NAS.

Summarized in Table 1, NAS-Perf-360 consists of problems that are conducive to processing by convolutional neural networks, which includes a trove of applications associated with spatial and temporal data, spanning single and multiple dimensions. Most current NAS methods are not implemented to search for other types of architectures to process tabular data and graph data. Therefore, we have set this scope for our investigation. During the selection of tasks, diversity is our primary consideration. We define the following axes of diversity to govern our task-filtering process: the first is problem dimensionality, including both 2D with matrix inputs and 1D with sequence inputs; the second is dataset size, for which our selection spans the scale from 1,000 to 1,000,000; the third is problem type , divisible into tasks requiring a singular prediction (point prediction) and multiple predictions (dense prediction); fourth and finally, diversity is achieved through selecting tasks from various learning objectives from applications of deep learning, where introducing NAS could improve upon the performance of handcrafted neural networks.

In lieu of providing raw data, we perform data pre-processing locally and store the processed data on a public Amazon Web Service's S3 data bucket with download links available on our website. Our data treatment largely follows the procedure defined by the researchers who provided them. This enhances reproducibility by ensuring the uniformity of input data for different pipelines. Specific pre-processing and augmentation steps are described below.

**CIFAR-100: Standard image classification**    As a starting point of comparison to existing benchmarks, we include the **CIFAR-100** task (Krizhevksy, 2009), which contains RGB images from natural settings to be classified into 100 fine-grained categories. CIFAR-100 is preferred over CIFAR-10 because it is more challenging and suffers less from over-fitting in previous research.

**Spherical: Classifying spherically projected CIFAR-100 images**    To test NAS methods applied to natural-image-like data, we consider the task of classifying spherical projections of the CIFAR-100 images, which we call **Spherical**. In addition to scientific interest, spherical image data is also present in various applications, such as omnidirectional vision in robotics and weather modeling in meteorology, as sensors usually produce distorted image signals in real-life settings. To create Spherical CIFAR, we project the planar signals of the CIFAR images to the northern hemisphere and add a random rotation to produce spherical signals for each channel following the procedure specified in Cohen et al. (2018). The resulting images are 60×60 pixels with RGB channels.

**NinaPro: Classifying electromyography signals**    **NinaPro** moves away from the image domain to classify hand gestures indicated by electromyography signals. For this, we use a subset of the NinaPro DB5 dataset (Atzori et al., 2012) in which two Myo armbands collect EMG signals from 10 test individuals who hold 18 different hand gestures to be classified. These armbands leverage data

Table 1: Information of tasks in NAS-Perf-360

| Task name | Size | Dim. | Type | Learning objective | New to NAS |
|-----------|------|------|------|--------------------|------------|
| CIFAR-100 | 60K | 2D | Point | Classify natural images into 100 classes | |
| Spherical | 60K | 2D | Point | Classify spherically projected images into 100 classes | ✓ |
| NinaPro | 3956 | 2D | Point | Classify sEMG signals into 18 classes corresponding to hand gestures | ✓ |
| FSD50K | 51K | 2D | Point (multi-label) | Classify sound events in log-mel spectrograms with 200 labels | ✓ |
| Darcy Flow | 1100 | 2D | Dense | Predict the final state of a fluid from its initial conditions | ✓ |
| PSICOV | 3606 | 2D | Dense | Predict pairwise distances between residuals from 2D protein sequence features | ✓ |
| Cosmic | 5250 | 2D | Dense | Predict propablistic maps to identify cosmic rays in telescope images | ✓ |
| ECG | 330K | 1D | Point | Detecting atrial cardiac disease from a ECG recording via classification | ✓ |
| Satellite | 1M | 1D | Point | Classify satellite image pixels' time series into 24 land cover types | ✓ |
| DeepSEA | 250K | 1D | Point (multi-label) | Predicting chromatin states and binding states of RNA-binding sequences | |

from muscle movement, which is collected using electrodes in the form of wave signals. Each wave signal is then sampled using a wavelength and frequency prescribed in Côté-Allard et al. (2019) to produce 2D signals.

**FSD50K: Labeling sound events**   FSD50K (Fonseca et al., 2020) is derived from the larger Freesound dataset (Fonseca et al., 2017) of Youtube videos with 51,000 clips totaling more than 100 hours of sound. These clips are manually labeled and equally distributed in 200 classes from the AudioSet ontology (Gemmeke et al., 2017). Each clip could receive multiple labels. Unlike TIMIT (Garofolo, 1993), FSD50K does not focus exclusively on sounds of spoken language but includes sound events from physical sources and production mechanisms. The mean average precision (mAP) is used to evaluate classification results.

**Darcy Flow: Solving partial differential equations (PDEs)**   Our first regression task, **Darcy Flow**, focuses on learning a map from the initial conditions of a PDE to the solution at a later timestep. This application aims to replace traditional solvers with learned neural networks, which can output a result in a single forward pass. The input is a 2d grid specifying the initial conditions of a fluid, and the output is a 2d grid specifying the fluid state at a later time, with the ground truth being the result computed by a traditional solver. We report the mean square error (MSE or $\ell_2$).

**PSICOV: Protein distance prediction**   PSICOV studies the use of neural networks in the protein folding prediction pipeline, which has recently received significant attention to the success of methods like AlphaFold (Jumper et al., 2020). While the dataset and method they use are too large-scale for our purposes, we consider a smaller set of protein structures to tackle the specific problem of inter-residual distance predictions outlined in Adhikari (2020b). 2D large-scale features are extracted from protein sequences, resulting in input feature maps with a massive number of channels. Correspondingly, the labels are pairwise-distance matrices with the same spatial dimension. The evaluation metric is mean absolute error (MAE or $\ell_1$) computed on distances below 8 Å, referred to as $MAE_8$.

**Cosmic: Identifying cosmic ray contamination**   Images from space-based facilities are prone to corruption by charged particles collectively referred to as "cosmic rays." Cosmic rays on images should be identified and masked before the images are used for further analysis (Zhang & Bloom,

Table 2: Performance of NAS and the WRN baselines across the tasks of NAS-Perf-360. Methods are divided into efficient methods (DenseNAS and fixed WRN) that take 1-10 GPU-hours, more expensive methods (DARTS and WRN tuned by ASHA) that take 10-100+ GPU-hours, and specialized methods (Auto-DL and AMBER). All results are averages of three random seeds.

| Search space | Search algorithm | CIFAR-100 0-1 error [l] | Spherical 0-1 error [l] | Darcy Flow relative $\ell_2$ [l] | PSICOV $MAE_8$ [l] | Cosmic FNR [l] |
|---|---|---|---|---|---|---|
| WRN | default | 23.35±0.05 | 85.77±0.71 | 0.073±0.001 | 3.84±0.053 | 51.76±2.09 |
| DenseNAS | random | 25.49±0.41 | 71.23±1.65 | 0.071±0.006 | 3.70±0.06 | 70.42±6.07 |
| DenseNAS | original | 25.98±0.38 | 72.99±0.95 | 0.10±0.01 | 3.84±0.15 | 79.52±2.20 |
| WRN | ASHA | 23.39±0.01 | 75.46±0.40 | 0.066±0.00 | 3.84±0.05 | 37.53±10.16 |
| DARTS | GAEA | 24.02±1.92 | 48.23±2.87 | 0.026±0.001 | 2.94±0.13 | 31.15±3.48 |
| Auto-DL | DARTS | n/a | n/a | 0.049±0.005 | 6.73±0.73 | 99.79±0.02 |
| Search space | Search algorithm | NinaPro 0-1 error [l] | FSD50K mAP [h] | ECG F1 score [h] | Satellite 0-1 error [l] | DeepSEA AUROC [h] |
| WRN | default | 6.78±0.26 | 0.08±0.001 | 0.57±0.01 | 15.49±0.03 | 0.60±0.001 |
| DenseNAS | random | 8.45±0.56 | 0.40±0.001 | 0.58±0.01 | 13.91±0.13 | 0.60±0.001 |
| DenseNAS | original | 10.17±1.31 | 0.36±0.002 | 0.60±0.01 | 13.81±0.69 | 0.60±0.001 |
| WRN | ASHA | 7.34±0.76 | 0.09±0.03 | 0.57±0.01 | 15.84±0.52 | 0.59±0.002 |
| DARTS | GAEA | 17.67±1.39 | 0.06±0.02 | 0.66±0.01 | 12.51±0.24 | 0.64±0.02 |
| AMBER | ENAS | n/a | n/a | 0.67±0.015 | 12.97±0.07 | 0.68±0.01 |

[h / l] a higher / lower value of the metric indicates better performance.

2020). The **Cosmic** task uses imaging data of local resolved galaxies collected from the Hubble Space Telescope. Inputs and outputs are same-size 2D matrices, with the output predicting whether each pixel in the input is an artifact of cosmic rays. We report the false-negative rate (FNR) of identification results.

**ECG: Detecting heart disease**    Electrocardiograms are frequently used in medicine to diagnose sinus rhythm irregularities. The **ECG** task is based on the 2017 PhysioNet Challenge (Clifford et al., 2017), with 9 to 60-second ECG recordings sampled at 300 Hz and labeled using four classes: normal, disease, other, or noisy rhythms. Recordings are processed using a fixed sliding window of 1,000 ms and stride of 500 ms. We report the F1-score according to the challenge's guidelines.

**Satellite: Satellite image time series analysis**    Satellite image time series (SITS) are becoming more widely available in earth monitoring applications. Our dataset comes from Formosat-2 satellite images acquired over Toulouse, France (Petitjean et al., 2012). Available in multiple channels, SITS track the land cover changes over several years as each pixel in the image represents a geographical region. The goal of the **Satellite** task is to generate land cover maps for geo-surveying. Specifically, a series of pixels in a given color channel constitute a time series to be classified into 46 land cover types.

**DeepSEA: Predicting functional effects from genetic sequences**    Predicting chromatin effects of genetic sequence alterations is a significant challenge in the field to understand genetic diseases. **DeepSEA** (Zhou & Troyanskaya, 2015), provides a compendium of genomic profiles from the Encyclopedia of DNA Elements (ENCODE) project (Consortium et al., 2004) to train a predictive model estimating the behavior of chromatin proteins, divided into 919 categories. Due to computation constraints, we subsample 36 of these categories as per Zhang et al. (2021a) and further take 5% of the training data for prediction. We report the area under the receiver operating characteristic (AUROC) following the previous work.

## 4    EXPERIMENTAL DESIGN

Having detailed our construction of NAS-Perf-360, we now describe a set of experiments to demonstrate its usefulness for evaluating NAS methods and guiding research on diverse tasks. In this section,

we first specify the different NAS methods and baselines we compare, followed by the experimental and reproducibility setup we follow. The resulting evaluations are reported in Table 2, aggregate performance in Table 3, and performance profiles in Figure 2.

## 4.1 BASELINES AND SEARCH PROCEDURES

As noted in Section 1, our initial experiments focus on two practitioners with different resource settings, one with enough compute to tune a WRN and another who can only train it once with the default hyperparameters. In matching these settings, we focus on two well-known search paradigms: cell-based NAS (using DARTS (Liu et al., 2019b)) and macro NAS (using DenseNAS (Fang et al., 2020)). We further compare these approaches to two customized NAS methods: Auto-DeepLab (Liu et al., 2019a) for 2D dense prediction and AMBER (Zhang et al., 2021b) for 1D prediction. We detail these approaches below.

**Wide ResNet with Hyperparameter Tuning**    Architectures based on ResNet He et al. (2016) are a common first choice by practitioners faced with a new domain (Fonseca et al., 2020; Adhikari, 2020b); it is thus a natural source of fixed-architecture baselines for our study. We use the Wide ResNet variant (Zagoruyko & Komodakis, 2016) with 16 layers and a widen factor of 4, and apply its original training routine directly for the constrained practitioner. For the other practitioner, we wrap the training procedure with a hyperparameter tuner, ASHA (Li et al., 2018), an asynchronous version of Hyperband (Li et al., 2017). Given a range for each hyperparameter, ASHA uniformly samples configurations and uses brackets of elimination: at each round, each configuration is trained for some epochs, before the algorithm selects the best-performing portion based on validation metrics. This procedure is useful for finding suitable hyperparameters in an easy-to-use, automated fashion.

**Cell-based Search Using DARTS**    The first NAS paradigm we consider is cell-based NAS. These methods first search for a genotype, a cell containing neural operations. During evaluation, an architecture is constructed by replicating the searched cell and stacking them together. The most popular search space for this approach is DARTS (Liu et al., 2019b), which assigns one of eight operations to six edges in two types of cells: "normal" cells preserve the shape of the input while "reduction" cells downsample it. For dense tasks, we do not use the reduction cell to prevent introducing a bottleneck. For 1D tasks, all convolutions in the cell are converted from 2D to 1D. Finally, to adhere to standard ML practices, we do *not* adapt the standard DARTS pipeline from the original paper. As this search space has been heavily studied, we use as a search routine a recent approach, GAEA PC-DARTS (GAEA), that achieves some of the best-known results on CIFAR-10 and ImageNet (Li et al., 2021a). This NAS approach, due to its heavy retraining routine, is compared to the tuned WRN baseline of the less-resource-constrained practitioner.

**Macro NAS Using DenseNAS**    The second NAS paradigm we consider is macro NAS. Instead of building from a fixed cell, it requires the specification of a supernet with different inter-connected blocks. These blocks and connections are then pruned to construct an architecture. For our benchmark, we choose a recent search space, DenseNAS (Fang et al., 2020), which has near state-of-the-art results on ImageNet. DenseNAS searches for architectures with densely-connected, customizable routing blocks to emulate DenseNet (Huang et al., 2017). In our experiments, we use the ResNet-based search space, DenseNAS-R1, which contains all neural operations of the WRN backbone. For 2D tasks, we adapt two super networks from the one used for ImageNet as inputs to the search algorithm. The super network for dense tasks maintains the same spatial dimensions without downsampling to avoid bottlenecks, and we lower the learning rate for evaluating architectures to prevent divergence. For transferring to 1D tasks, all network operations are switched from 2D to 1D. Other training and evaluation procedures are identical to those in the original paper and uniform across all tasks. DenseNAS is quick to search and evaluate, making it comparable to the fixed WRN baseline.

We apply another search method to the fixed DenseNAS space to study the relative importance of search algorithms. Random search is implemented through randomly sampling architectures from the DenseNAS space and validating them after a brief training period of 10 epochs before evaluating the best performer. To ensure fairness of comparison, we allot equal GPU hours to random search and regular DenseNAS search, additionally applying a soft constraint that random architecture model sizes should not surpass DenseNAS searched architecture sizes significantly.

Table 3: Median rank and performance improvement over WRN across NAS-Perf-360.

| Search space
Search algorithm | WRN
default | DenseNAS
original | DenseNAS
random | WRN
ASHA | DARTS
GAEA | Auto-DL
DARTS | AMBER
ENAS |
|---|---|---|---|---|---|---|---|
| Median rank | 4.0 | 4.0 | 4.0 | 3.5 | 1.5 | $6.0^{\dagger}$ | $1.0^{\dagger}$ |
| % better than WRN* | 0.0% | 2.53% | 0.0% | 0.0% | 20.1% | $-75.3\%^{\dagger}$ | $20.0\%^{\dagger}$ |

* relative improvement over the default (untuned) WRN baseline
† metric computed only on the subset of three tasks on which the method was evaluated

**Domain-specific NAS Baselines: Auto-DL and AMBER**   To study the importance of search spaces, we further handpick two domain-specific NAS approaches applicable only to a subset of the tasks. Using an encoder-decoder architecture, Auto-DeepLab (Auto-DL) (Liu et al., 2019a) is designed for dense prediction, e.g., semantic segmentation. While the decoder is fixed, Auto-DL searches for an encoder via first-order DARTS. We evaluate Auto-DL on Darcy Flow, PSICOV, and Cosmic tasks.

AMBER (Zhang et al., 2021b) aims to automate neural network design for 1D genomic data. This framework establishes a 12-layer network backbone and parametrizes a long-short term memory network (LSTM) as a controller to search for suitable 1D operations and residual connections, following the ENAS (Pham et al., 2018) optimization protocol. At each step, the controller samples architectures to compute reward based on area under the receiver operating characteristics (AUROC) before outputting the highest-reward architecture. We evaluate AMBER on the ECG, Satellite, and DeepSEA tasks.

## 4.2 EXPERIMENTAL SETUP

Below we discuss the main reporting details of our empirical evaluation.

- **Hyperparameter tuning:** As detailed in the Appendix, we use the same hyperparameter ranges across all tasks to tune WRN. The tuning budget is selected to match that of DARTS (GAEA).
- **Aggregation Metrics:** Table 3 contains the median rank and relative improvement over WRN of each method for direct comparison via a single number. We also employ performance profiles (Dolan & Moré, 2002) in Figure 2, an approach that allows an analysis taking into account outliers while not allowing small differences in performance to dominate; as described in Figure 1 these curves denote for each $\tau$ the fraction of tasks on which a method is no worse than a $\tau$-factor from the optimal.
- **Software and hardware:** We adopt the free, open-source software *Determined*[2] for experiment management, hyperparameter tuning, and cloud deployment. All experiments are performed on a single p3.2xlarge instance with an NVIDIA V100 GPU. Costs in GPU-hours are in the appendix.

## 5 DISCUSSION

We conclude our presentation of NAS-Perf-360 via an analysis that (a) reveals new insights about the capabilities and robustness of current NAS methods and (b) demonstrates how our benchmark can enable critical next steps in NAS research. In particular, we start by considering our two practitioners faced with a choice of spending their limited compute on a (possibly tuned) fixed-architecture CNN or trying to find a better architecture using NAS. With this study, we investigate whether modern NAS methods perform well beyond the tasks for which they were designed.

1. A surface-level analysis suggests that under light resource constraints, modern NAS in the form of DARTS (GAEA) is quite robust to a wide variety of tasks: Table 3 shows it is the highest-ranked domain-independent method and attains the most significant improvement over the fixed WRN baseline. The performance profile in Figure 2 (left) also seems favorable, although it is overtaken by tuned WRN at a higher $\tau$-suboptimality. However, a closer look at 2D point tasks in Figure 2 (right) reveals that DARTS is quite poor there, despite its design domain being image classification; in particular, it performs very poorly on NinaPro and FSD50K. Furthermore, on tasks where it performs well, it can still lag behind expert architectures; for example, on Darcy Flow, networks that use FNO (Li et al., 2021b) or XD (Roberts et al., 2021) operations do much

---

[2]GitHub repository: `https://github.com/determined-ai/determined`

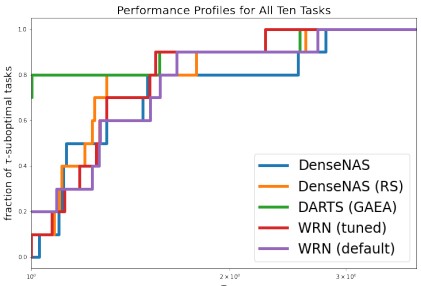 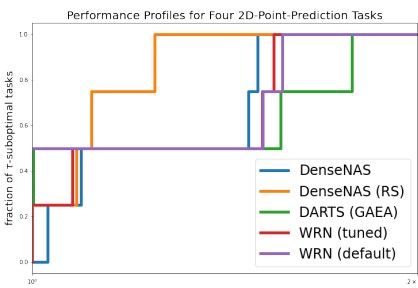

Figure 2: Performance profiles on all tasks (left) and 2D point tasks (right). The y-value indicates the fraction of tasks on which a plotted method's error is within a multiplicative factor $\tau$ of the lowest error achieved by all plotted methods..

better. Overall, our results suggest that this practitioner can apply NAS and expect to see some improvement, but also risks catastrophically poor performance (e.g., FSD50K) or not getting truly state-of-the-art results (e.g., Darcy Flow).

2. Under stronger budget constraints, our experiments strongly suggest that a practitioner should simply apply the default WRN to their problem rather than undergo the additional complexity of using DenseNAS, as the latter attains little-to-no improvement over the former in Table 3 and has a usually-worse performance profiles Figure 2. One bright point, however, is the strong performance of DenseNAS on FSD50K, where it outperforms all methods even while DARTS (GAEA) fails.

These first experiments suggest that the modern NAS methods are not always robust to diverse tasks, especially under resource-constrained settings. We believe that NAS-Perf-360's main roles as a future benchmark include developing an understanding of the multi-domain performance of existing approaches and guiding research into better NAS methods. While the latter is beyond the scope of this paper, our additional experiments demonstrate how NAS-Perf-360 facilitates the former.

Notably, several results address the question of the relative importance of search space v. search algorithm. For example, Table 3 shows that on DenseNAS, random search is nearly identical to the more sophisticated weight-sharing scheme of the original paper; the two algorithms' performance profiles are also difficult to distinguish in Figure 2. Furthermore, AMBER—a 1D NAS method whose search space includes larger-kernel convolutions for handling such tasks—does better than GAEA even though it uses an older search algorithm (ENAS). These both suggest that search space design, including the use of a wider variety of operations, may be at least as crucial for success as the search algorithm. This point is reinforced by example tasks such as Darcy Flow, where architectures with more exotic operations substantially outperform our best results, as discussed earlier.

NAS-Perf-360 also reveals failure points of several methods, not just general ones that usually perform quite well such as DARTS (GAEA) but also the objective-specific approach Auto-DL, which despite being designed for dense prediction tasks, does poorly on all those considered here. Understanding when and why these performance drops happen is critical to developing a more robust NAS that is useful not just on average but in more challenging settings.

## 6 CONCLUSION

NAS-Perf-360 is a new performance benchmark consisting of ten diverse tasks derived from various fields of research and practice. It is designed for reproducible research on an academic budget that will guide the development of NAS methods and other automated approaches towards more robust performance across different domains. In initial results, we have demonstrated both the need for such a benchmark and the utility of NAS-Perf-360 specifically for developing new search spaces and algorithms; we welcome researchers to use its tasks to develop new procedures for automating ML.

## 7 ETHICS STATEMENT

Within our array of tasks, only NinaPro, ECG, and DeepSEA contain human-derived data. Our chosen subset of NinaPro contains only muscle movement data without any exposure of personal information. The original experiments to acquire NinaPro data are approved by the ethics commission of the canton of Valais, Switzerland (Atzori et al., 2012). The ECG data derives from an open challenge and is provided by the medical device company AliveCor, under the GPL license allowing it for public use. The DeepSEA data derived from ENCODE is part of an international collaborative effort, which is overseen and funded by the National Human Genome Research Institute (NHGRI). For other datasets, we have listed the data licenses in the appendix. While we do not view the specific datasets in NAS-Perf-360 as potential candidates for misuse, the broader goal of applying NAS to new domains comes with inherent risks that may require mitigation on an application-by-application basis.

## 8 REPRODUCIBILITY STATEMENT

The following measures are taken to ensure reproducibility:

1. We store the processed data in AWS S3, and data splits are the same for all experiments.
2. Code is always executed in a fixed Docker container using a pre-built image on Docker Hub. This guarantees a uniform execution environment and saves users from configuring dependencies.
3. Via the specification of a random seed, *Determined* controls several essential sources of randomness during execution, including hyperparameter sampling and training data shuffling.
4. During training, we validate on the entire validation set, not on a mini-batch, to limit stochasticity.
5. Code and download links of all datasets are available at the anonymized repository: `https://anonymous.4open.science/r/NAS-Bench-360-26D1`

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
