# OpenReview forum: "NAS-Bench-360: Benchmarking Diverse Tasks for Neural Architecture Search"
_ICLR.cc/2022/Conference — ICLR 2022 Submitted_

### Official Review · Reviewer_zB8j · 2021-11-02

**Correctness:** 3
**Technical Novelty And Significance:** 4
**Empirical Novelty And Significance:** 2
**Recommendation:** 5
**Confidence:** 4

**Main Review:**

Strength:

1/ I think the problem this paper raises is very meaningful. Currently, NAS has been over-focused on a few problems, especially, image classification on ImageNet or CIFAR10 datasets. It is not clear wether progress achieved on NAS will transfer to other problems and applications.

2/ The NAS benchmark proposed in this paper is flexible and general enough to include architecture search, hyper-parameter tuning, and different base architectures. This is better than previous NAS Benchmarks, where the scope is limited to cell-level architecture space on a specific problem. Also, the computational cost for this NAS benchmark is much more manageable.

3/ The conclusion or take-away message of this paper is useful for practitioners. Especially, depending on the computational budgets, whether to choose to use an off-the-shelf model with HP tuning or to search architectures.

Weaknesses:
1/ Despite that authors clarified their principles of choosing different tasks in this benchmark, the exact choice is still quite arbitrary. Eventually, the reason to have a benchmark is that 1) the benchmark itself measure some metrics we care about, or 2) performance on this benchmark can generalize to a wider range of tasks that may arise in practical applications. In this paper, scenario-1) obviously does not hold. But for scenario-2), this has not been demonstrated in the paper. Specifically, does a NAS method that performs well on this benchmark can generalize better to new unseen tasks? There is no experimental or theoretical support for this, which limits the contribution of this work.

2/ The technical contribution of this paper is limited. This paper did not propose new methods or other novel contributions. An example of a more significant contribution might be to propose a better NAS algorithm that generalizes better than existing ones.

3/ The paper is not well written and organized. For example, the metrics in Figure 1 is not clearly explained. The caption explains the metric as "A larger value indicates a larger fraction of tasks on which the method is within a multiplicative factor \tau of the best." The question is, the best of what? The best performance in terms of accuracy or error rate? What is "multiplicative factor \tau of the best"? I don't think this is clearly explained anywhere in the paper, which makes the results confusing.

**Summary Of The Paper:**

This paper proposes a benchmark to test the performance of NAS algorithms and search spaces on a diverse set of tasks. The benchmark consists of 10 different datasets across different modalities. On these tasks, a variety of NAS algorithm as well as search spaces are allowed a fixed amount of compute resources (in terms of GPU hours) to explore and train, and reach the final performance. The search space are not limited to architecture topologies, but also hyper-parameters. On this new benchmark, authors find that existing SoTA NAS methods may not generalize to different tasks, especially with low compute budgets.

**Summary Of The Review:**

The paper proposes a meaningful direction and benchmark for NAS. But whether this benchmark can measure the generalization of NAS method is not clear, the technical contribution of the paper is limited, and the writing of the paper should be improved.

---

> ### Author Response · Authors · 2021-11-19
> **Generalization studied in this work; our technical contribution explained**
>
> Thank you for your detailed critique of the work. Below we respond to your critique:
>
> [*Specifically, does a NAS method that performs well on this benchmark can generalize better to new unseen tasks? There is no experimental or theoretical support for this, which limits the contribution of this work.*]
>
> Instead of choosing arbitrary datasets, we have selected many candidates initially and filtered out the suite used in NAS-Perf-360, which are diverse individually but share similarities as shown in Table 1. Therefore, we think that the generalization performance of NAS methods can be studied by comparing performance across similar tasks within the benchmark. One possible grouping of similar tasks is 2D point prediction (4 tasks), 2D dense prediction (3 tasks), and 1D prediction (3 tasks).
> Within each group, our empirical results provide insights into how well domain-specific NAS methods generalize. For instance, for 1D prediction, AMBER is originally designed on DeepSEA but shows generalization by beating other NAS methods and the baseline WRN on Satellite and ECG, which are “unseen” tasks. In contrast, AutoDeepLab performs well on image segmentation datasets as claimed by its authors, but it fails in comparison to other methods on our subgroup of 2D dense prediction tasks. We believe that these performance insights can also be helpful for practitioners when choosing which NAS method to apply.
>
> [*The technical contribution of this paper is limited. This paper did not propose new methods or other novel contributions.*]
>
> The goal of this paper is to construct a useful benchmark for NAS, not a new NAS method. Furthermore, we do contribute technical novelty by successfully transferring NAS methods designed for 2D data to 1D tasks. A practitioner cannot simply feed 1D data to their original implementations. To accomplish this transfer, we designed new DARTS-like and DenseNAS-like search spaces with 1D neural operations and adapted the optimization pipeline for these modified spaces. At the time of this paper’s writing, we were not aware of any other works that implemented a 1D adaptation of DARTS or DenseNAS. Therefore, we introduce these new, adapted search spaces and implementations as part of the contribution besides the empirical results. Unfortunately, due to space limitations, we did not discuss this technical contribution in detail, only briefly mentioning them in Section 4.1.
>
>
> [*The paper is not well written and organized. For example, the metrics in Figure 1 is not clearly explained. The caption explains the metric as "A larger value indicates a larger fraction of tasks on which the method is within a multiplicative factor \tau of the best." The question is, the best of what? The best performance in terms of accuracy or error rate? What is "multiplicative factor \tau of the best"?*]
>
> We apologize for the confusion on this caption. We are using the error rate as the metric, and “the best” refers to the lowest error rate achieved by all plotted methods on an individual task. We will change the caption in the revision to: “the y-value indicates the fraction of tasks on which a plotted method’s error is within a multiplicative factor \tau of the lowest error achieved by all plotted methods.” Do you have other suggestions on the writing and organization? The other reviewers seem to find the paper clear and well-written (reviewer 8TV7), though we realize that these opinions are subjective.

---

> > ### Comment · Reviewer_zB8j · 2021-11-22
> > **Reply to authors.**
> >
> > Thanks the authors for their response.
> > - Generalization of this benchmark.
> >
> > The authors argue that "the generalization performance of NAS methods can be studied by comparing performance across similar tasks within the benchmark". This is sensible, but not sufficient. I recognize the value of building a benchmark that inspires the community to build NAS algorithms that can generalize. It will be great if a practitioner can look at this benchmark and pick the best NAS method and apply to a new problem. I think the benchmark in its current form is not yet sufficient to deliver the goal. The number of tasks contained in this benchmark is too small and there is no strong evidence (empirical or theoretical) to support that a good performance on this benchmark will transfer well to new tasks. I think there are still a lot of work to be done for this paper to realize its potential and provide value to the community. Therefore, I stand with my earlier decision on this paper.
> >
> > - Technical novelty.
> >
> > Thanks authors for noting their adaptation of NAS methods to 1D. I recognize this as an non-trivial effort to enable the experiments and claims of the paper, but I don't think it is a ground-breaking novel contribution. Therefore, my judgement on the paper's novelty contribution stays the same.

---

### Official Review · Reviewer_JYCo · 2021-11-03

**Correctness:** 3
**Technical Novelty And Significance:** 2
**Empirical Novelty And Significance:** 3
**Recommendation:** 5
**Confidence:** 4

**Main Review:**

+ The attempt of creating new benchmarks or datasets is very promising for the NAS field.
+ The proposed benchmark is much diverse compared with the existing ones.

Concerns:
- The attempt is important but this paper seems to be just providing the existing datasets. For NAS evaluation, it is also important to provide the rank of each candidate architecture in the search space because recent studies show that NAS methods find architectures achieving good performance, but their ranks in the search space are far from the best [Yu+, ICLR'20]. It indicates that it is necessary to provide the rank or something related to the rank to appropriately evaluate NAS methods, not only performance.

- Depending on a given task, an optimal search space would change. But this paper uses the limited search space for the tasks, thus I am wondering if this benchmark can appropriately evaluate NAS methods. It seems that designing search space itself is needed for diverse tasks and evaluation of NAS methods on them.

- It would be better to describe the difference between the existing NAS benchmarks and the proposed one in more detail.
- It would be nice to provide the detail of the search space in the main paper.

**Summary Of The Paper:**

This paper proposes a new benchmark for NAS methods, which is called NAS-Bench-360. Unlike the existing benchmark datasets for NAS, the proposed benchmark contains ten diverse tasks derived from various fields of research. This paper has tested several standard NAS methods on the proposed benchmark and confirmed that there are many gaps among the ten tasks and NAS methods.

**Summary Of The Review:**

The attempt of creating a new benchmark is very important and this paper shows promising results. However, my major concern is about the design of the proposed benchmark (see concerns above). Hopefully, the authors can address my concern in the rebuttal period.

---

> ### Author Response · Authors · 2021-11-19
> **Rename to NAS-Perf-360; more information on search spaces used in this work**
>
> [*The attempt is important but this paper seems to be just providing the existing datasets. For NAS evaluation, it is also important to provide the rank of each candidate architecture in the search space because recent studies show that NAS methods find architectures achieving good performance, but their ranks in the search space are far from the best [Yu+, ICLR'20]. It indicates that it is necessary to provide the rank or something related to the rank to appropriately evaluate NAS methods, not only performance.*]
>
> To clarify, we are not providing a tabular NAS benchmark in this work, and we will change its name to NAS-Perf-360 as noted in the AC discussion. The rank of each architecture is only obtainable in a tabular benchmark after evaluating each architecture, which is unavailable in our case. The search spaces that we consider, such as DARTS with 10^18 architectures, are so large in size that no group can evaluate them completely.
>
> Note also that the rank of an architecture is a relative metric, not an absolute one like accuracy. Even if an architecture is the best in a search space, it is possible that this architecture is bad if the space itself contains only weak architectures. This case is exemplified as our architecture searched on DARTS beats the best architecture in the NAS-Bench-201 search space on CIFAR-100 accuracy. For these gigantic search spaces, we think performance is the preferred absolute metric.
>
> [*Depending on a given task, an optimal search space would change. But this paper uses the limited search space for the tasks, thus I am wondering if this benchmark can appropriately evaluate NAS methods. It seems that designing search space itself is needed for diverse tasks and evaluation of NAS methods on them.*]
>
> In this work, we designed new DARTS-like and DenseNAS-like search spaces with 1D neural operations, which is briefly described in Section 4.1, so we did not use the same search space for all tasks. However, we disagree that different tasks require different search spaces, as designing a new search space for each application domain goes against the goal of automation. What we did was using the same search space for a family of similar tasks, such as 1D prediction or 2D dense prediction as classified in Table 1.
>
> [*It would be better to describe the difference between the existing NAS benchmarks and the proposed one in more detail.*]
>
> Now that we have renamed our submission to a non-benchmark name, it is not meant to be compared to the existing NAS benchmarks. Existing benchmarks provide evaluation results of every single architecture on one or few datasets, whereas NAS-Perf-360 benchmarks methods by their performance on a diverse set of datasets.
>
> [*It would be nice to provide the detail of the search space in the main paper.*]
>
> We have evaluated a total of four search space families in this work instead of one single search space. Due to space constraints, we cannot describe each in detail except for providing overviews in Section 4.1. Maybe these pictorial representations of the research space can better explain them rather than a long description:
> - DARTS: [example cells searched on CIFAR-10](https://drive.google.com/file/d/1dkfdTBQjhVYMvxmKUHSxNz-XMLFajlpU/view?usp=sharing)
> - DenseNAS: [an example architecture searched on ImageNet](https://drive.google.com/file/d/1FZ28rGJMMOd1wKziZ66altYJe3_g0aDj/view?usp=sharing)
> - AMBER: [search space consisting of computational operations - blue box and residual connections - pink box](https://drive.google.com/file/d/1CtZ0Uj8YKTa48k1AmeH_XNlahmLph4t7/view?usp=sharing)
> - AutoDeepLab: [an example search space](https://drive.google.com/file/d/1KPz4IIffGTiVTS7s_vZB6sPgMVpp5-Lb/view?usp=sharing)

---

> > ### Comment · Reviewer_JYCo · 2021-11-22
> > **Reply to the authors**
> >
> > - change its name to NAS-Perf-360
> >
> > If this paper does not provide the rank of each architecture (i.e. tabular NAS benchmark), the contribution of this paper is limited since this work seems to be just evaluating some standard NAS algorithms on various datasets and providing some empirical results. It would be better to provide further insights and deep analysis on the proposed benchmarks.
> >
> > - evaluation metric and search space
> >
> > Although the authors argue that designing a new search space for each application domain goes against the goal of automation, I disagree with this point and the design of a search space is also important for NAS evaluation. As the search space has a large impact on the performance of NAS algorithms, a benchmark should provide enough search spaces for evaluation. Otherwise, it will be hard to tell which a NAS algorithm itself or a search space affect the final performance. However, it doesn't seem that this work design a search space for each dataset carefully.
> >
> > Also, the ranking of each architecture should be used for evaluation as well as accuracy. As recent studies e.g., [Yu+, ICLR'20] pointed out, it is difficult to judge whether a NAS algorithm works well or not only from accuracy.
> >
> > [Yu+, ICLR'20] Evaluating the Search Phase of Neural Architecture Search, ICLR'20.

---

### Official Review · Reviewer_ofGn · 2021-11-03

**Correctness:** 4
**Technical Novelty And Significance:** 2
**Empirical Novelty And Significance:** 2
**Recommendation:** 3
**Confidence:** 5

**Main Review:**

In this work, the authors thrown out an issue that the existing NAS benchmarks focused on the popular tasks -- image classification and needs more diverse tasks for the NAS evaluation. First of all, the existing NAS benchmarks have covered at least 6 different tasks/datasets and if we counting the dataset/tasks explored in the HPO works (which also try to find some architecture options), we have more architecture datasets/benchmarks in the NAS community, which is strong and diverse. So that some claims in the paper may be exaggerated.

As for the proposed "benchmark", one important feature for the NAS benchmark is the associated architecture datasets, which provides tabuler or suggrate performance data for the architectures. Unfortunately, this work does not provide such architecture dataset.

As for the collected tasks, some may be too simple and not suitable for evaluating the DL or NAS methods. For example, some datasets only has about 1K data but try to do some 2D dense prediction. It is not supuring some approaches, which require enough data, perform poorly on such dataset.

From the technical part, the authors compared a few NAS method and basic DL models. As the NAS techniques have been applied in many many different domains/tasks. If we collect the results from the papers that use DARTS in a non-CIFAR/classification application, the effect seems to be the similar. Compared to this option, the only benefit of this NAS-Bench-360 would be the calibration the NAS methods in the existing application papers.

**Summary Of The Paper:**

The authors argued that existing NAS benchmarks targets the well-studied tasks, and thus proposed that we should evaluate NAS on diverse tasks. And thus the authors collected 10 tasks and did some analysis on these ten tasks/datasets to compare several NAS methods and some strong baseline DL models, such as wide resnet.

**Summary Of The Review:**

This NAS benchmark lacks architecture datasets for the collected ten datasets and technical novelty.

---

> ### Author Response · Authors · 2021-11-19
> **Rename to NAS-Perf-360; the value of diverse tasks in this work**
>
> Thank you for your review. Following our discussion with the AC, we have renamed our submission to NAS-Perf-360 to distinguish our work from typical tabular and surrogate NAS benchmarks.
>
> Instead of an architecture dataset, our contribution is a dataset of datasets for NAS. Regarding the specific merits of this dataset and its member tasks, we have some clarifications to address your concerns:
>
> [ *an issue that the existing NAS benchmarks focused on the popular tasks -- image classification*]
>
> Our motivational problem is the lack of NAS studies outside of popular domains, which are computer vision and NLP. We are conscious of the usage of tasks beyond image classification, but out-of-domain tasks remain understudied. In the specific case of NAS-HPO-Bench, its datasets are drawn from the standard UCI dataset repertoire and does not overlap with our efforts to curate realistic tasks for NAS.
>
>
>
> [*some may be too simple and not suitable for evaluating the DL or NAS methods”*]
>
> We disagree with this statement, as by observing the decent performance of hand-designed neural networks (Table 5, Appendix) we have demonstrated the viability of using deep learning on these tasks, for both large and small datasets. We think the study of NAS in data-scarce settings is also valuable. NAS does not always underperform with less data as shown in PSICOV, and 1K data in Darcy Flow is “enough” for training since FNO networks achieve a very low error rate.

---

> > ### Comment · Reviewer_ofGn · 2021-11-29
> > **New Name Is Good; Still have other concerns**
> >
> > Dear Authors,
> >
> > Thanks for your responses. The new name looks good to me.
> > However, my major concerns are that there are a lot (not lack) of NAS studies outside of CV/NLP. For example, there are a lot of NAS works studying speech tasks, RL-related tasks, time-series tasks, etc. At least, there work evaluated the random search algorithm and DARTS algorithm. Based on these literature works, NAS-Perf-360 are highly overlapped with them and less comprehensive than them (if we consider them as a whole). Therefore, I would like to keep my rating.
> >
> > Best regards,
> > Reviewer ofGn

---

### Official Review · Reviewer_8TV7 · 2021-11-04

**Correctness:** 4
**Technical Novelty And Significance:** 1
**Empirical Novelty And Significance:** 3
**Recommendation:** 6
**Confidence:** 5

**Details Of Ethics Concerns:**

The paper contains some datasets that involve human subjects. This is clearly stated by the authors in Section 7.

**Main Review:**

In general this paper is really well-written and the motivation behind its idea is quite valid in my opinion. In general I think the NAS community will benefit from this suite of different tasks to run their NAS methods. Below I list some more details comments:

**Pros:**

(+) I think the set of collected tasks will be a useful contribution to the NAS community in order to shift the attention to other tasks that do not only involve image classification.

(+) Useful insights on the usefulness of NAS by evaluating different methods or/and search space across 10 different diverse tasks.

(+) The paper is well-structured and easy to follow.

(+) Available codebase that allows reproducing the experiments and evaluating new methods by using the same settings.

**Cons:**

(-) I think the paper could benefit more by experiments that provide further insights for the NAS practitioner. For instance, performance predictors are commonly used nowadays in NAS [1]. It would be useful to evaluate a couple of these methods (e.g. some of the ones used in [1]) on the proposed suite, both in isolation (reporting the rank correlation coefficient) and inside a black-box algorithm (e.g. Bayesian optimization).

(-) I think there is some text and results in the main paper that can be moved to the appendix and make some space for additional empirical results. For instance, the authors can move Table 3 to the appendix, since it is somewhat redundant with Table 2. Some examples of such additional experiments can be for instance:
- A more detailed investigation on why the NAS methods fail in many of the tasks in NAS-Bench-360;
- Why some of these NAS methods perform better in some tasks and some others not? Is this because of a sub-optimal search space (operation choices and topology) or an underperforming NAS optimizer?
- How much does the inductive bias affect the ranking of different NAS methods evaluated throughout the 10 tasks? By NAS methods I mean: Different NAS optimizers evaluated on a fixed search space.

**Other comments:**

- I think the term "NAS-Bench" is reserved for tabular/surrogate benchmarks and using it for this submission might be confusing at first for many readers, since NAS-Bench-360 in principle is only a collection of datasets and scripts for running methods (search space + NAS algorithm) on them. I would suggest the authors to reconsider changing the name of their suite.

- I would also add plots showing the performance over time of the different evaluated NAS methods on every task.

- I agree with the authors that as an AutoML practitioner, in general I am also concerned which will provide the best output: 1) optimizing the architecture when keeping the hyperparameters fixed or 2) the hyperparameters of a fixed well-performing architecture. However, one can also do both, i.e. given a fixed computational budget, start with 1) and then move to 2), or the other way around. It would be interesting to see this experiment with a NAS search space that includes the WRN architecture (does the DenseNAS space include WRN? If yes, one can very well use that.).

- Why Table 5 is in the Appendix and not in the main paper instead of Table 2?

**References**

[1] White et al. How powerful are performance predictors in neural architecture search?, In NeurIPS 2021

**Summary Of The Paper:**

This paper presents NAS-Bench-360, which consists of collection of 10 diverse tasks, carefully selected to include datasets with different number of classes, total number of datapoints, dimensionality and application domain. By using a fair evaluation protocol, the authors benchmark different NAS methods on various search spaces, with various degrees of inductive bias, and provide some interesting insights on the effectiveness of NAS under time constraints, showing that there is almost no benefit compared to using a fixed manually engineered architecture.

**Summary Of The Review:**

I really like the contributions and the insights offered by this paper. Even though there is no novelty per se, I think the NAS community will benefit from this curated collection of diverse tasks. I lean towards acceptance and I will increase my score after most of my concerns are addressed.

---

> ### Author Response · Authors · 2021-11-19
> **General Response to Reviewer 8TV7**
>
> Thank you for being supportive. Here we hope to address some of your concerns:
>
> [*It would be useful to evaluate a couple of [performance prediction] methods*]
>
> Thank you for the suggestion. We considered evaluating performance prediction methods on the DARTS search space but found that using them did not make sense in the setting we considered. In particular, the vast majority of the time spent in the DARTS pipeline is in the retraining phase, so taking a more heuristic approach in the search phase would not save enough time in the entire pipeline to make it worth the loss in accuracy.
>
> [*I think there is some text and results in the main paper that can be moved to the appendix and make some space for additional empirical results. For instance, the authors can move Table 3 to the appendix, since it is somewhat redundant with Table 2.  Some examples of such additional experiments can be for instance:
> A more detailed investigation on why the NAS methods fail in many of the tasks in NAS-Bench-360;
> Why some of these NAS methods perform better in some tasks and some others not? Is this because of a sub-optimal search space (operation choices and topology) or an underperforming NAS optimizer?
> How much does the inductive bias affect the ranking of different NAS methods evaluated throughout the 10 tasks? By NAS methods I mean: Different NAS optimizers evaluated on a fixed search space.*]
>
> Part of the goal of Table 3 is to succinctly present results from additional experimental results. For example, in an effort to study your first two empirical suggestions, we checked whether the DenseNAS optimizer was causing its poor performance and found that in many cases simple random search does just as well on that search space. As for the last question, it’s unclear to us what an “inductive bias” is when discussing a NAS method.
>
> [*I think the term "NAS-Bench" is reserved for tabular/surrogate benchmarks and using it for this submission might be confusing at first for many readers, since NAS-Bench-360 in principle is only a collection of datasets and scripts for running methods (search space + NAS algorithm) on them. I would suggest the authors to reconsider changing the name of their suite.*]
>
> As noted in the AC discussion, we will change it to NAS-Perf-360.
>
> [*I would also add plots showing the performance over time of the different evaluated NAS methods on every task.*]
>
> Thanks for the suggestion. We can’t rerun the models and generate these plots at the moment, but we will perform a final run in a month or so that we can add these plots in the camera-ready version.
>
> [*I agree with the authors that as an AutoML practitioner, in general I am also concerned which will provide the best output: 1) optimizing the architecture when keeping the hyperparameters fixed or 2) the hyperparameters of a fixed well-performing architecture. However, one can also do both, i.e. given a fixed computational budget, start with 1) and then move to 2), or the other way around. It would be interesting to see this experiment with a NAS search space that includes the WRN architecture (does the DenseNAS space include WRN? If yes, one can very well use that.).*]
>
> We agree that a practical approach would be to tune hyperparameters for the specific architecture. However, this approach has not been addressed by our methods, or even by anyone else.
>
> [*Why Table 5 is in the Appendix and not in the main paper instead of Table 2?*]
>
> Apart from the content that is also in Table 2, the content in Table 5 contains information that is more succinctly summarized by Table 3.
>
> In general, we agree with your assessment that our work could benefit from more experimentation studying different NAS methods. However, due to the cost of these NAS evaluations, which takes hundreds of GPU hours, we are unable to generate these results during the discussion period. The experiments will appear in the camera-ready version.

---

> > ### Comment · Reviewer_8TV7 · 2021-11-22
> > **Reply to authors**
> >
> > Thank you very much for your clarifications. Below I provide more details on what I meant exactly with my previous comments.
> >
> > **"It would be useful to evaluate a couple of performance prediction methods."**
> >
> > I think this is indeed a useful experiment considering that performance predictors, such as model-based ones (GPs, XGBoost, GNNs) or zero-cost proxies [[Abdelfattah et al. 2021](https://arxiv.org/abs/2101.08134)], have been of wide interest throughout the community recently as another efficient alternative to weight-sharing NAS. Please check [White et al. 2021](https://arxiv.org/abs/2104.01177) for an comprehensive empirical evaluation of such methods. Without knowing the exact amount of compute power that is available to the authors, my idea was to sample a certain number of architectures uniformly at random for each (or some) of the search space and on every of the considered tasks. Then split these in train/test and fit such performance predictors on the training set and compute some metric, e.g. the rank correlation, on the test set. This is also important when considering one-shot methods such as DARTS, which can be used as performance predictors via evaluating sampled architectures using the shared weights.
> >
> > **"It’s unclear to us what an “inductive bias” is when discussing a NAS method."**
> >
> > I apologize for not being clearer about this. What I meant was how much important is having a more specialized search space tailored to the problem at hand, e.g. AutoDeepLab for dense prediction tasks. It seems from Table 2 that for the 3 dense prediction tasks considered it does not perform as well as the other methods. Is this because of the small number of training examples in these datasets? Or is it because of the time allocated for optimization?
> >
> > **"We can’t rerun the models and generate these plots at the moment..."**
> >
> > What I meant is to add the incumbent trajectories for black-box optimizers such as random search and ASHA. For the weight-sharing methods this can be expensive, but still it would be interesting to see if there is any overfitting on the validation set, as it is not uncommon for methods such as DARTS [[Zela et al.](https://openreview.net/forum?id=H1gDNyrKDS)].
> >
> > **"We agree that a practical approach would be to tune hyperparameters for the specific architecture. However, this approach has not been addressed by our methods, or even by anyone else."**
> >
> > I still think this can be a nice addition to the experiments that the authors run in their paper. To be clearer, this is the setup I had in mind:
> > Given a fixed time budget *K* for optimizing (either architecture or other hyperparameters) do:
> > - Keep architecture fixed and optimize only hyperparameters during the whole budget *K*.
> > - Keep hyperparameters fixed and optimize only the architecture during the whole budget *K*.
> > - Start with a good default fixed architecture and tune hyperparameters for half the budget *K/2* and then take the optimal hyperparameters and optimize the architecture for the rest of the budget *K/2*.
> > - Start with good default hyperparameters and tune architecture for half the budget *K/2* and then take the optimal architecture and optimize the hyperparameters for the rest of the budget *K/2*.
> >
> > Thanks again for your reply and for your contribution to the NAS community. I will keep my acceptance score.

---

### Official Review · Reviewer_oZbZ · 2021-11-05

**Correctness:** 3
**Technical Novelty And Significance:** 1
**Empirical Novelty And Significance:** 1
**Recommendation:** 3
**Confidence:** 5

**Main Review:**

This paper proposes a 'so-called' NAS-Bench-360, which is quite misleading. The original NASBench idea (brought by NASBench-101 Ying et al.) is to exhaustively train all (or a substantial amount of) architectures within a search space and record their training / evaluation statistics for researcher or NAS practitioners to evaluate their search algorithm in a fair manner. However, this work mis-use the concept. After reading the entire paper for multiple times, it seems this work is merely evaluating some standard NAS algorithms on various datasets and present some empricial analysis.

In my humble opinion, the usefulness of NAS-Bench is to leverage some groups' massive compute-power to obtain a massive pool of architecture statistics on various tasks, to benefit people without such power to conduct NAS research. Evaluting some algorithms, even across drastically different tasks, is interesting but not enough to be accepted to a top tier conference.



**Summary Of The Paper:**

This paper proposes to evaluate some standard NAS algorithms over various tasks, ranging from image classification, sattllite image time series analysis or ECG detecting hard diseases.



**Summary Of The Review:**

See above, this paper mis-use the concept of NAS-Bench that is common to NAS researcher. After reading the entire paper, I do not find it interesting as a NAS practitioner. Evaluating some algorithms on different tasks is a good empirical report but cannot be qualified as a top-tier conference paper due to lack of novelty and effort.

---

> ### Author Response · Authors · 2021-11-19
> **Rename to NAS-Perf-360; contributions as a dataset of datasets**
>
> Thank you for your direct review. We are renaming the paper as “NAS-Perf-360” for this submission to avoid confusion and distinguish our work from tabular benchmarks. We hope that resolving this naming issue can address part of your concerns.
>
> More importantly, we would like to highlight this work’s contribution of a dataset of datasets for NAS similar to *meta-dataset* for meta-learning. Our dataset comprises a carefully curated suite of tasks spanning many diversity axes as shown in Table 1, and these (except CIFAR-100) are not off-the-shelf like CIFAR-10 or ImageNet. This dataset could be useful for the NAS community to verify the empirical generalizability of new and existing methods. We think the dataset is novel within the field. As per the message from the AC, datasets are within the scope of this conference
>
> It is on top of this dataset for datasets that we evaluated the algorithms and provided reproducible benchmarking results, drawing attention to the failure cases of NAS. We hope that you could reassess the effort behind this work, accounting for not only the empirical studies but also the data curation process.

---

### Author Response · Authors · 2021-11-15
**A General Response to Reviewers**


Thank you for your constructive reviews. We respectfully disagree that NAS benchmarks can only be tabular. Like an ML benchmark (such as [MLPerf](https://mlcommons.org/en/training-normal-10/)), NAS-Bench-360 evaluates and critiques existing methods on many datasets, and more importantly provides fully reproducible experiment procedures and data for community use. A tabular NAS benchmark is usually restricted to one problem domain and one (often small) search space, and our non-tabular benchmark is its complement by addressing these limitations and redirecting attention to understudied tasks. What we compromise in depth we gain in breadth. Boundary-drawing on the concept of the NAS benchmark alienates alternative contributions and implicitly narrows the definition of benchmarking to evaluate all the architectures. What if the search space is too large to be fully evaluated (e.g. [DARTS](https://arxiv.org/abs/1806.09055)), and what if the search space has an infinite number of architectures (e.g. [XD-Operations](https://arxiv.org/abs/2103.15798))? The infeasibility of tabular architecture datasets in these scenarios calls for other forms of benchmarks to study these search spaces.

We also argue that datasets should not be automatically considered substandard for top-tier venues such as ICLR. Datasets fuel algorithmic developments and are valuable contributions in and of themselves as resources in ML. In meta-learning, [Meta-Dataset](https://arxiv.org/abs/1903.03096) has been published at this conference and inspired follow-up works using it for empirical research. The suite of datasets in our work aims to achieve a similar goal in NAS to advocate for evaluations on more realistic tasks, and the dataset entails a nontrivial effort to organize and preprocess data from various sources. Finally, our dataset achieves novelty because it provides a framework for systematic comparison of NAS methods across tasks. Overall, from the perspectives of value, effort, and novelty, good datasets should not be denied publication at top venues.

---

### Author Response · Authors · 2021-11-21
**Revision posted**

We thank the reviewers for their valuable feedback. To reflect the name change and clarify some captions, we have posted a revision of the paper.

---

### Decision · Program_Chairs · 2022-01-20

**Decision:**

Reject

**Comment:**

This paper at first used the name NAS-Bench-360 for the benchmark, which confused several reviewers (who expected a tabular benchmark behind this name). The authors renamed the benchmark, which removed this issue, emphasizing that the contribution does not lie in proposed a new tabular NAS benchmark, but a new performance evaluation of NAS on different data sets.
One reviewer recommended acceptance, but 3 reviewers stuck with their rejection scores, the reasons being that
- there are by now several papers applying NAS outside of computer vision, with a seemingly more comprehensive analysis
- more analysis would be useful
- it is unclear how general the conclusions are that can be drawn from performance on the included datasets.
(Low technical novelty was also mentioned, but I do not believe that this type of paper can be very impactful even if it has no technical novelty.)

Overall, although I agree with the accepting reviewer that this type of work can be very useful to the community, the rejecting reviewers have too many criticisms to accept the paper in its current form. I encourage the authors to address them and to resubmit.
One note (which did not affect the decision, but which I'd like to notify the authors about) is that a reviewer found that the author identity was revealed in the anonymous codes provided by the authors (https://anonymous.4open.science/r/NAS-Bench-360-26D1).